# 1,25-Dihydroxyvitamin D3 and 20-Hydroxyvitamin D3 Upregulate LAIR-1 and Attenuate Collagen Induced Arthritis

**DOI:** 10.3390/ijms222413342

**Published:** 2021-12-12

**Authors:** Linda K. Myers, Michael Winstead, John D. Kee, Jeoungeun J. Park, Sicheng Zhang, Wei Li, Ae-Kyung Yi, John M. Stuart, Edward F. Rosloniec, David D. Brand, Robert C. Tuckey, Andrzej T. Slominski, Arnold E. Postlethwaite, Andrew H. Kang

**Affiliations:** 1Department of Pediatrics, University of Tennessee Health Science Center, 50 N. Dunlap, Rm. 461R, Memphis, TN 38103, USA; 2Department of Medicine, University of Tennessee Health Science Center, 956 Court Ave., Memphis, TN 38163, USA; winstead.michael@gmail.com (M.W.); aszehri@wakehealth.edu (J.D.K.); jeoungeunpark27@gmail.com (J.J.P.); jstuart45@gmail.com (J.M.S.); acody@uthsc.edu (A.E.P.); akang@uthsc.edu (A.H.K.); 3Department of Pharmaceutical Sciences, University of Tennessee Health Science Center, 881 Madison Ave, Memphis, TN 38103, USA; szhang71@uthsc.edu (S.Z.); wli@uthsc.edu (W.L.); 4Department of Microbiology-Immunology-Biochemistry, University of Tennessee Health Science Center, 858 Madison Ave., Memphis, TN 38163, USA; ayi@uthsc.edu; 5Memphis Veterans Affairs Medical Center, 1030 Jefferson Ave., Memphis, TN 38104, USA; erosloni@uthsc.edu (E.F.R.); dbrand@uthsc.edu (D.D.B.); 6School of Molecular Sciences, University of Western Australia, 35 Stirling Highway, Perth, WA 6009, Australia; robert.tuckey@uwa.edu.au; 7Department of Dermatology, University of Alabama at Birmingham 500 22nd St. S, Birmingham, AL 35294, USA; aslominski@uabmc.edu; 8Comprehensive Cancer Center, University of Alabama at Birmingham 1824 6th Ave., Birmingham, AL 35294, USA; 9Birmingham Veterans Affairs Medical Center, 700 19th Street S., Birmingham, AL 35233, USA

**Keywords:** arthritis, vitamin D, 20*S*(OH)D3, LAIR-1, autoimmunity

## Abstract

Vitamin D plays a crucial role in regulation of the immune response. However, treatment of autoimmune diseases with 1,25-dihydroxyvitamin D3 [1,25(OH)_2_D3] doses sufficient to be effective is prohibitive due to its calcemic and toxic effects. We use the collagen-induced arthritis (CIA) model to analyze the efficacy of the noncalcemic analog of vitamin D, 20*S*-hydroxyvitamin D3 [20*S*(OH)D3], as well as 1,25(OH)_2_D3, to attenuate arthritis and explore a potential mechanism of action. Mice fed a diet deficient in vitamin D developed a more severe arthritis characterized by enhanced secretion of T cell inflammatory cytokines, compared to mice fed a normal diet. The T cell inflammatory cytokines were effectively suppressed, however, by culture of the cells with 20*S*(OH)D3. Interestingly, one of the consequences of culture with 1,25(OH)_2_D3 or 20*S*(OH)D3, was upregulation of the natural inhibitory receptor leukocyte associated immunoglobulin-like receptor-1 (LAIR-1 or CD305). Polyclonal antibodies which activate LAIR-1 were also capable of attenuating arthritis. Moreover, oral therapy with active forms of vitamin D suppressed arthritis in LAIR-1 sufficient DR1 mice, but were ineffective in LAIR-1^−/−^ deficient mice. Taken together, these data show that the effect of vitamin D on inflammation is at least, in part, mediated by LAIR-1 and that non-calcemic 20*S*(OH)D3 may be a promising therapeutic agent for the treatment of autoimmune diseases such as Rheumatoid Arthritis.

## 1. Introduction

Human autoimmune arthritis causes significant joint damage due to dysregulated autoimmunity. The risk and progression of Rheumatoid Arthritis (RA) have been correlated with vitamin D deficiency (low serum 25-hydroxyvitamin D [25(OH)D]). However, attempts to treat RA with 1,25(OH)_2_D3 and its precursors, 25(OH)D3 have caused hypercalcemic toxicity when given chronically at the pharmacological doses needed to maximally suppress arthritis and autoimmunity [1]. This side effect limits the amounts that can be given chronically to patients with autoimmune diseases such as RA.

We have discovered a novel pathway of D3 metabolism operative in humans, initiated by cytochrome P450scc (CYP11A1), and with subsequent involvement of CYP27B1, which generates additional biologically active products [2,3,4,5]. These are at least as potent as classical 1,25(OH)_2_D3 when tested in vitro and in vivo in several model systems and, like 1,25(OH)_2_D3, bind to the vitamin D receptor (VDR) [6,7,8,9,10,11,12,13]. A major product of this pathway, 20*S*(OH)D3, is nontoxic (i.e., noncalcemic with limited effects on the hematopoietic system, liver, kidney, and heart) at doses as high as 60 µg/kg, while 25(OH)D3 or 1,25(OH)_2_D3 induce hypercalcemia at doses ≤2 μg/kg [14,15,16,17]. Importantly, 20*S*(OH)D3 has recently been found to be present in all 103 human serum samples analyzed, with a mean concentration of 0.27 ng/mL, which is 5 times higher than the mean concentration of 1,25(OH)_2_D3 in these samples [4,13,18]. Thus, 20*S*(OH)D3 is a natural product that has also been detected in honey [12].

The binding of 1,25(OH)_2_D3 to the intracellular VDR regulates multiple genes involved in many physiological processes [19]. In fact, the human genome contains over 23,000 VDR binding sites, most of which are cell-specific [20]. As such, vitamin D is now considered essential for the maintenance of physiological homeostasis. Its deficiency has been associated with a wide range of diseases and cardiovascular and metabolic disorders, including cancer, hypertension, and infectious and autoimmune diseases. The wide variety of vitamin D effects on the immune response suggests that vitamin D may hold therapeutic promise in many autoimmune diseases. In RA patients, serum 25(OH)D3 levels correlated negatively with disease activity [21], and each 10 ng/mL increase in serum 25(OH)D3 was associated with a 0.3-point decrease in mean DAS28 and a 25% decrease in serum C-reactive protein [21]. Moreover, in interventional trials, higher doses of supplementation with 1,25(OH)_2_D3 have been associated with decreased pain and significant declines in C-reactive protein levels together with a trend toward clinical efficacy [22,23,24,25].

We recently reported that 20(OH)D3 markedly reduces clinical signs of arthritis and joint damage in a mouse model of RA, by suppressing immune responses by T and B cells [26]. These results were related to reduction in CD4^+^ T cells, CD19^+^ B cells, anti-CII antibodies, and maintenance of CD4^+^CD24^+^FoxP3^+^ Tregs. In the current study, we have used the collagen-induced arthritis (CIA) model to investigate the effects of 1,25(OH)_2_D3 and non-calcemic 20*S*(OH)D3 with results revealing a new mechanism involving LAIR-1 by which they modulate T-cell function, and ultimately, autoimmune arthritis.

## 2. Results

### 2.1. Murine Autoimmune Arthritis and Vitamin D Decificency

Although vitamin D is known to play a critical role in calcium homeostasis, it also modulates the immune system [27]. To confirm that vitamin D plays a role in autoimmune arthritis, groups of DBA/1 mice were fed diets either deficient (VitD−) or sufficient in vitamin D (VitD+) beginning on day 21 when they were weaned. When mice were immunized with CII/CFA to induce arthritis, those fed diets deficient in vitamin D had significantly more severe arthritis than those fed a normal diet (Figure 1). The cytokine data in Table 1 support the arthritis data. When splenocytes from mice fed diets either vitamin D sufficient (D+) or vitamin D deficient (D−) were cultured with the immunodominant A2 peptide, cells from mice fed with the vitamin D deficient diet secreted greater T cell cytokine responses than those from mice fed the vitamin D sufficient diet. Although multiple CD4^+^ T cell lineages were affected, the inflammatory Th1 and Th17 responses were most pronounced, creating a relative shift toward an inflammatory profile (Table 1). 

### 2.2. Cytokine Responses Using 20S(OH)D3

Since new safer noncalcemic secosteroids have been developed [28], it was important to test their effectiveness in attenuating in vitro murine T cell inflammatory responses. To this end, 20*S*(OH)D3 was selected. Splenocytes from DBA^qCII24^ mice were cultured with the immunodominant peptide A2 alone or in the presence of 20*S*(OH)D3. The resulting supernatants were evaluated for cytokine responses. As shown in Figure 2, 20*S*(OH)D3 significantly inhibited secretion of inflammatory cytokines (IFNγ and IL-17), while it increased the IL-4 and IL-10. The net result was a relative shift towards a Th2, non-inflammatory phenotype. The increase in Th2 cytokine responses induced by treatment with 20(OH)D3 is similar to observations made by others following treatment with 1,25(OH)2D3 [29], yet differs from the downregulation of all T cell cytokines noted in Table 1 in cells from mice given vitamin D sufficient diets. Although both conditions induce a relative shift away from an inflammatory T cell profile, diets sufficient in vitamin D expose the entire immune system of the animal to vitamin D, modifying a large number of T cells over several months, while treatment of normal cells with 20(OH)D3 in vitro reflects a much shorter exposure. We believe these varying conditions, as well as differences in the shifting B/T cell rations that vitamin D induces explain the differing results. These studies support our premise that 20*S*(OH)D3 will be an effective therapeutic alternative in autoimmunity [16].

### 2.3. Vitamin D Can Upregulate LAIR-1

One effective way to downregulate the inflammatory immune response is to activate inhibitory receptors. Since the expression of inhibitory receptors varies depending upon the activation or differentiation state of the cell during autoimmunity [30], it was of interest to determine how culture with the analog of vitamin D3 affected receptor expression. Selecting the natural inhibitory receptor leukocyte associated immunoglobulin-like receptor-1 (LAIR-1) for further testing, human PBMCs were cultured overnight with either 1,25(OH)_2_D3, 20*S*(OH)D3, or α1(II), the purified constituent polypeptide chain of type II collagen. When the cultured cells were lysed and analyzed by Western blotting using an anti- LAIR-1 antibody (Figure 3), results showed that LAIR-1 expression was appreciably increased by culture with either of the two active forms of vitamin D3, as well as α1(II), a natural ligand of LAIR-1. In a similar fashion, surface LAIR-1 on murine CD4+ cells was upregulated by culture with 1,25(OH)_2_D3 (Figure 3, lower panel). Taken together, these results demonstrate that LAIR-1 activation could help explain the resulting suppression of inflammation by 1,25(OH)_2_D3 and CYP11A1-derived 20*S*(OH)D3.

### 2.4. LAIR-1 Also Suppresses Arthritis

Since polyclonal anti-LAIR-1 antibodies predominately activate LAIR-1, we used the collagen-induced arthritis model to induce in vivo activation of LAIR-1 and examined the resulting autoimmune arthritis. Groups of mice were immunized with CII/CFA to induce arthritis, then were given interperitoneally either polyclonal anti-LAIR-1 antibodies or rabbit IgG. When mice were scored for severity of arthritis (Figure 4), as hypothesized, mice treated with α-LAIR-1 antibodies had significantly less severe arthritis than mice given the control IgG. These data confirm that activation of LAIR-1 can downregulate autoimmune arthritis.

### 2.5. Vitamin D Treatment in CIA Using LAIR-^−/−^ and LAIR-1^+/+^ Mice

To confirm that the suppressive effects of 1,25(OH)_2_D3 and 20*S*(OH)D3 were mediated by LAIR-1, both LAIR-1^+/+^ and LAIR-1^−/−^ mice were immunized with CII/CFA to induce arthritis followed by treatment orally with either vehicle control or 1,25(OH)_2_D3. When the mean severity of arthritis scores for each group were calculated, the wild-type mice that received 1,25(OH)_2_D3 supplementation had a significant suppression of arthritis, having the lowest severity scores of any group (*p* < 0.05) compared to vehicle controls. Importantly, the data also show that LAIR-1^−/−^ mice had severity scores that were not suppressed by 1,25(OH)_2_D3 treatment. An additional five LAIR-1^−/−^ mice were treated by oral gavage with 20S(OH)D3 from days 13 to 44. Although these doses suppressed CIA in LAIR^+/+^ mice, the LAIR^−/−^ mice fed 20S(OH)D3 had severity scores similar to the LAIR^−/−^ vehicle controls (Figure 5, lower panel). Moreover, 20S(OH)D3, given at lower doses than 1,25(OH)2D3, decreased severity scores more potently than 1,25(OH)2D3. Taken together these data reveal that LAIR-1 is critical for the attenuation of inflammation induced by treatment with 1,25(OH)_2_D3 or 20*S*(OH)D3.

## 3. Discussion

To understand the mechanisms by which 1,25(OH)_2_D3 and its non-calcemic analog 20*S*(OH)D3 modulate T cell function and ultimately autoimmune arthritis, pre-clinical studies were performed using the collagen induced arthritis (CIA) model. CIA is an autoimmune arthritis model induced in genetically susceptible mice that leads to synovitis and both cartilage and bone erosion, similar to what is seen in human RA. Our data show that mice fed a diet deficient in vitamin D develop a more severe arthritis. Moreover, the mechanism for inhibiting inflammatory T cell cytokines and autoimmune arthritis with vitamin D involves the inhibitory receptor LAIR-1. These studies suggest a new mechanism by which active forms of vitamin D effectively modulate autoimmune arthritis.

Vitamin D is an extremely important current public health issue. In recent years, reports of potential health benefits of vitamin D outside its well-known role in bone and mineral metabolism, coupled with some evidence that portions of the population may be vitamin D-deficient, have generated both interest and controversy over the proper vitamin D target levels and how best to achieve them [31,32,33]. Although the most well-known function of vitamin D is in maintaining the right balance between calcium and phosphate serum levels, thus promoting bone health, other functions have been identified [19].

Considering that earlier studies demonstrated that 1,25(OH)_2_D3 inhibited arthritis in the CIA model in mice fed a low-calcium diet to protect against development of hypercalcemia [34], it is possible that vitamin D supplementation may have beneficial effects for patients with RA [35,36,37]. Unfortunately, supraphysiological doses of vitamin D3 would be required to obtain therapeutic concentrations in vivo. Increasing the vitamin D intake to diminish the incidence and severity of diseases such as RA, type 1 diabetes, IBD, and MS, and to obtain immunomodulating effects in vitro requires local concentrations of 1,25(OH)_2_D3 of about 10^−10^ M, which may be associated with an unacceptable level of hypercalcemia.

Several natural vitamin D analogs, including 20*S*(OH)D3, have been discovered which do not induce hypercalcemia and may have great therapeutic value [16,26]. The discovery of new secosteroidogenic pathways initiated by the action of CP11A1 led to the characterization of 20*S*(OH)D3 which has potent anti-proliferative and prodifferentiation effects. It has fewer side effects than other active vitamin D compounds because it is non-calcemic at concentrations as high as 60 µg/kg [17]. Importantly, 20*S*(OH)D3 and its direct metabolite 20*S*,23*S*(OH)_2_D3 act as potent inhibitors of NF-κB [9,28], RORα, and potentially RORγt [38,39]. Furthermore, since 20*S*(OH)D3 is produced in vivo by mitochondria in adrenal glands and other steroidogenic tissues [2,3], it is a natural product, even detectable in honey [12].

Vitamin D regulates both innate and adaptive immunity. Previous studies that examined the direct effect of vitamin D3 on T cells generally found it to be inhibitory for pro-inflammatory Th1 and Th17 cytokine production. Active forms of vitamin D primarily utilize the VDR (an intracellular transcription factor) to regulate gene expression in T cells [40]. Since CD4^+^ T cells from VDR^−/−^ mice produce more IFNγ and less IL-4 and IL-5 compared to T cells from wild-type mice [41], it is believed that the addition of 1,25(OH)_2_D3 inhibits the Th1 and Th17 response by acting in the nucleus to inhibit RORγt and T-bet while upregulating GATA-3 [42,43,44]. In cell culture experiments, 1,25(OH)_2_D3 can induce Tregs in the presence of IL-2, by upregulating the transcription of Fox-P3 and CTLA4 [45,46]. We now propose a third mechanism, the upregulation of the inhibitory receptor LAIR-1.

Our data demonstrate a novel observation, that both 1,25(OH)_2_D3 and its non-calcemic analog 20S(OH)D upregulate LAIR-1 on T cells. We propose that the ability to upregulate LAIR-1 on lymphocytes is a critical step in its efficacy. Previous studies that examined the direct effect of 1,25(OH)_2_D3 on T cells generally found it to inhibit pro-inflammatory Th1 and Th17 cytokine production [47,48,49]. This effect can now be explained in part by upregulation of the inhibitory receptor LAIR-1.

LAIR-1 belongs to a family of immune inhibitory receptors that protect against autoimmune dysfunction and tissue damage [50,51]. Activation of the receptor inhibits many cellular processes that are important to properly functioning inflammatory cells [52,53]. Subsequently, loss of LAIR-1 expression from host inflammatory cell surfaces allows immune-mediated damage of the synovium. In this study, we show that upregulating natural inhibitory receptors presents a method for suppressing autoimmune arthritis. Their immunoreceptor tyrosine-based inhibition motif (ITIM) acts as a negative regulator of immune cell receptor signaling. In our previous studies, we found that LAIR-1 engagement by alpha chains of collagen or C1q [54] led to inhibition of TCR signaling and decreased activation levels of key components of the canonical T cell signaling pathway, including Lck, Lyn, Zap-70, and the three MAP Kinases (ERK1/2, JNK1/2 and p38). Given the vital role of LAIR-1 in mitigating the autoimmune destruction of host cells, it seems plausible that upregulation of these receptors can play a major role in mitigating autoimmune inflammation, possibly providing a new immunotherapeutic target which downregulates the T-cell cytokine response. 

Limitations of this study: Our study highlights a potential role for 20S(OH)D3 in the treatment of autoimmune arthritis and demonstrates the importance of LAIR-1 in its mechanism of suppression. Our data are limited by the number of mice analyzed and treated with 20(OH)D3. We understand that studies using murine models of arthritis do not always reflect patient outcomes so we hope these studies will lead the way for future clinical trials. 

## 4. Materials and Methods

### 4.1. Animals

DBA/1 mice were obtained from the Jackson Laboratories (Bar Harbor, ME, USA) and B6 mice expressing the chimeric (human/mouse) DRB1*0101 construct were obtained from Taconic Biosciences (Hudson, NY, USA). The chimeric DRB1*0101 construct has been previously described, as has the production of Tg mice expressing this construct [55]. Mice transgenic for a CII-specific TCR-Vα11.1/Vβ8.3 having a DBA/1 background, referred to as DBA^qCII24^ [56] were developed and bred in the animal core facility of the Rheumatic Diseases Research Core Center, University of Tennessee Health Science Center as described previously [57].

LAIR-1 KO (knockout) mice [58] were crossbred to B6.DR1 transgenic mice with a B6 background for 12 generations. Genomic DNA was obtained from blood samples and PCR used to identify mice homozygous for either the LAIR-1^−/−^ or LAIR^+/+^ and expressing the DR1 transgene.

Mice were fed standard rodent chow (Ralston Purina Co., St. Louis, MO, USA) and water ad libitum. Sentinel mice were routinely tested for a panel of mouse pathogens. All animals were kept until the age of 7–10 weeks before being used for experiments. In some experiments, DBA/1 mice were fed either a vitamin D deficient (D−) or a vitamin D sufficient (D+) control diet (Teklad Diets, Harlan Envigo, Indianapolis, IN, USA) beginning at three weeks of age. Animal care and housing requirements set forth by the National Institutes of Health Committee on Care and Use of Laboratory Animals of the Institute of Laboratory Animal Resources were followed, and animal protocols were reviewed and approved by the Animal Care and Use Committees of the University of Tennessee Health Science Center (UTHSC) and the Memphis VA Medical Center. 

### 4.2. Type I and Type II Collagen

Native type II collagen (CII) was solubilized from fetal calf articular cartilage and native type I collagen (CI) from bovine hides by limited pepsin-digestion and purification as described earlier [59]. The purified collagen was dissolved in cold 10 mM acetic acid at 4 mg/mL and stored frozen at −80 °C until used. In some experiments type I collage from Advanced Biomatrix (Carlsbad, CA, USA) was used. α1(II) and α1(I) represent the constituent protein chains of bovine CII and CI respectively, isolated by carboxymethyl-cellulose chromatography. The synthetic peptides were supplied by Biomolecules Midwest Inc. (Waterloo, IL, USA). A peptide representing the immunodominant determinant of CII (GIAGFKGEQGPKGEB) (IEDBID 109115), is referred to as A2 or wild type (WT).

### 4.3. Production and Purification of 20S(OH)D3

20*S*(OH)D3 was originally reported by using enzymatic hydroxylation of D3 catalyzed by CYP11A1 as previously described by our group [2,15,60]. We have subsequently developed synthetic chemistry methods for large-scale production and demonstrated that the chemically synthesized 20*S*(OH)D3 is identical to the biologically generated 20*S*(OH)D3 [61,62]. In this study, both biochemically and chemically generated 20S(OH)D3 were used in the experiments. For biochemical production of 20*S*(OH)D3, 10 mmol/L VitD3 in 45% 2-hydroxpropyl-β-cyclodextrin was prepared. A buffer was prepared containing 20 mmol/L HEPES (pH7.4), 100 mmol/L NaCl, 0.1 mmol/L dithiothreitol, 2 µmol/L human cytochrome P450 scc, 0.1 mM EDTA, 0.3 µM adrenodoxin reductase, 10 µmol/L adrenodoxin, 2 mmol/L glucose 6-phosphate, 2 U/mL glucose 6-phosphate dehydrogenase, and 50 µmol/L NADPH. Then, 12.5 mL of this buffer were mixed with 200 µL of the VitD3 stock solution giving a final VitD3 concentration of 200 µmol/L and 0.9% concentration of 2-hydroxypropyl-β-cyclodextrin. After an 8 min pre-incubation, the reactions were initiated by adding NADPH, after which the samples were incubated for 3 h at 37 °C with slow shaking. After the 3 h incubation, 20 mL of ice cold dichloromethane were added to stop the reactions, after which the reaction products were extracted with dichloromethane as previously described [60,63,64]. Chemical production of 20*S*(OH)D3 were performed following our well-established procedures [62]. Final purification of 20*S*(OH)D3 was carried out using preparative thin-layer chromatography, followed by preparative reverse phase high-performance liquid chromatography as previously described [64]. Aliquots of the purified 20*S*(OH)D3 were lyophilized and stored at −80 °C until used.

### 4.4. Immunizations and Arthritis Induction

Six-to-8-wk-old mice were immunized with CII for the induction of arthritis. CII was dissolved in 10 mM cold acetic acid and emulsified at a 1:1 (*v*/*v*) ratio with complete Freund’s Adjuvant (CFA) containing 4 mg/mL of *M*. tuberculosis strain H37 Ra (Difco Microbiology Products, Becton Dickinson, NJ, USA) as previously described [59]. Mice were immunized subcutaneously at the base of the tail with 100 µg of CII. In some experiments, mice were given intraperitoneal injections on day 0, 7, and 14 with either 100 µg/dose of α-LAIR polyclonal IgG antibodies (Invitrogen, Waltham, MA, USA) or 100 µg/dose of normal rabbit IgG (Cell Signaling Technology, Inc., Beverly, MA, USA) as a control. The ant-LAIR-1 polyclonal antibodies were developed by immunizing rabbits with a recombinant protein corresponding to murine LAIR-1 and purifying with antigen affinity chromatograph, Protein A. The polyclonal α-Lair-1 sera was tested on murine CD4+ T cells to confirm upregulation of surface LAIR-1 using flow cytometry. In other experiments mice were treated by oral gavage with either control (propylene glycol, 0.1 mL/dose) or 1,25(OH)_2_D3, (25 μg/dose) or 15 µg/kg/dose 20*S*(OH)D3. In one experiment, the oral gavage was given three times a week, beginning the day after immunization of mice with CII/CFA and continuing for the duration of the experiment. In another experiment the oral gavage was given daily beginning day 13 after immunization and continuing to day 44.

The severity of arthritis was determined by visually examining each forepaw and hindpaw and scoring them on a scale of 0–4 as described previously [59]. Scoring was conducted by two examiners, one of whom was unaware of the identity of the treatment groups. Each mouse was scored thrice weekly beginning three weeks post immunization and continuing for 8 weeks. The mean severity score (sum of the severity scores for the group on each day /total number of animals in the group) was recorded at each time point. 

#### Preparation of Human PBMCs

Heparinized blood was obtained and diluted 1:3 with RPMI 1640 (#61870036) containing penicillin (100 µ/mL) and streptomycin (100 µg/mL) with 10% fetal calf serum (Thermo-Fischer Scientific, Memphis, TN, USA). The PBMC were isolated by Ficoll-Hypaque (Fischer Scientific, Memphis, TN, USA) by gently layering the diluted serum over an equal volume of Ficoll in a Falcon tube and centrifuging for 30–40 min at 400–500× *g* without a brake. The PBMCs were set up in culture overnight with vehicle control (ethanol, 10^−8^ mol/L), 1,25(OH)_2_D3 (10^−8^ mol/L), 20*S*(OH)D3 (10^−8^ mol/L), or αI(II) (100 µg/mL) overnight.

This study was conducted and approved by the University of Tennessee Health Science Center (UTHSC) at the Memphis Institutional Review Board. The Declaration of Helsinki protocols were followed and normal healthy donors gave their written informed consent. Adults (ages 18–70 years) were used in this study.

### 4.5. Cytokines

Splenocytes from DBA/1 mice which had been previously immunized with CII/CFA were cultured with A2 peptide (3 µmol/mL) or no Ag (*n* = 3 for each group). In some experiments, the cells were taken from mice that were either vitamin D sufficient or vitamin D deficient. In other experiments culture was done in the presence of 20*S*(OH) D3 (10^−7^ M). The experiments were run in triplicate and 72 h later supernatants were analyzed for the presence of IL-10, IL-4, IFN-γ, and IL-17A. The fluorescence was measured with a Bio-Plex MAGPIX Multiplex Reader (Bio-Rad, Hercules, CA, USA). Values are expressed as picograms per ml and represent the mean values for each group taken from three separate experiments.

### 4.6. Analysis of Protein Phosphorylation

Proteins in whole cell lysates were separated using SDS-PAGE gels and electrotransferred onto nitrocellulose membranes. After transfer, the membrane was blocked in tris buffered saline (TBS)-Tween 20 containing 5% BSA for 1 h and incubated 2 h with phospho- specific antibodies. The membrane was then incubated with a secondary antibody (Bio-Rad) for 1 hand subjected to Enhanced Chemiluminescence detection (ECL Western Blot Substrate, Pierce) according to the manufacturer’s protocol. To detect protein levels, the membranes were re-stripped and reblotted with non-phospho-specific antibodies.

### 4.7. Flow Cytometric Assessment of Lymphoid Cells

Murine spleens or lymph node cells were collected and the phenotype was determined by multiparameter flow cytometry using an SORP BD LSRII flow cytometer (BD Biosciences, San Jose, CA, USA). Some experiments were done following culture overnight with 1,25(OH)_2_D3 (10^−8^ M). Cells were labeled with fluorochrome antibodies including: PE (phycoerythrin)-conjugated anti-LAIR-1and FITC (fluorescein isothiocyanate)-conjugated anti-CD4. All were used according to the manufacturer’s recommendations. A minimum of 10,000 cells was analyzed from each sample and the final analysis was performed using FlowJo software (Tree Star, Ashland, OR, USA).

### 4.8. Statistical Analysis

Calculations to determine statistical significance of the tests were carried out using the programs SAS and GraphPad Prism 4. Depending on the data, a one-way ANOVA or student’s *t* test analysis was performed. Comparison of mean variable values with a distribution significantly different from normal in two unrelated groups was performed using the Mann–Whitney test, while in more than two unrelated groups using the Kruskal–Wallis test. *p* < 0.05 was considered statistically significant and *p* values are indicated in the figure legends.

## 5. Conclusions

We have performed pre-clinical studies using the collagen-induced arthritis (CIA) model. These reveal a new mechanism involving upregulating the expression of the inhibitory receptor LAIR-1 by which active forms of vitamin D effectively modulate autoimmune arthritis. Inflammatory cytokines from T cells as well as inflammation of autoimmune arthritis are attenuated by upregulation of the LAIR-1. The noncalcemic 20*S*(OH)D3 is as effective and less toxic than the classical form of vitamin D_3_ [1,25(OH)_2_D3]. These data will provide the basis for further research trials using non-calcemic analogs of vitamin D therapeutically to treat arthritis.

## Figures and Tables

**Figure 1 ijms-22-13342-f001:**
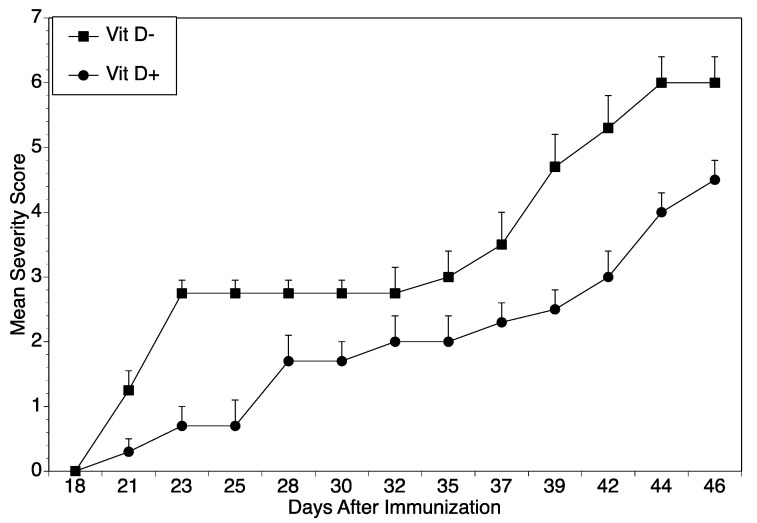
Disease severity in mice fed a vitamin D deficient diet. DBA/1 mice were fed either a vitamin D deficient (Vit D−) or a vitamin D sufficient (VitD+) control diet beginning at three weeks of age. All animals were challenged with a dose of 100 µg of CII emulsified in CFA for the induction of disease. Mice were scored for arthritis severity (*n* = 6 per group). The severity scores are expressed as means ± SEM. Treated mice fed the Vit D− diet were significantly different from controls beginning at day 39 (*p* ≤ 0.05 using Mann and Whitney analysis).

**Figure 2 ijms-22-13342-f002:**
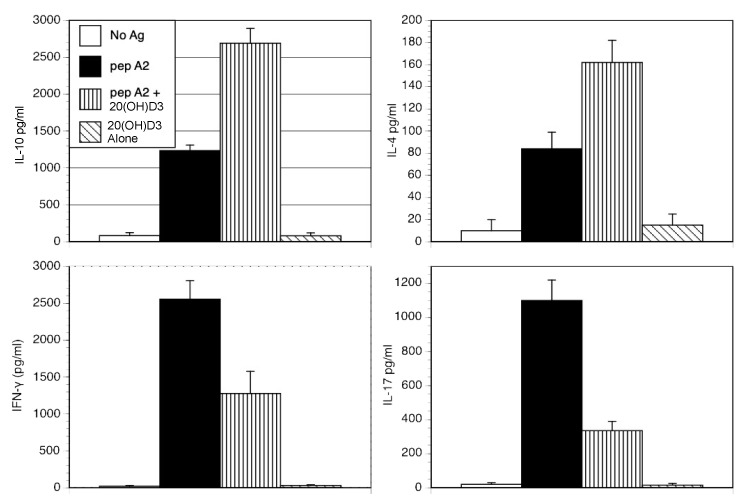
20S(OH)D3 modulates production of T cell cytokines. DBA/1^qCII^ splenocytes were cultured with the immunodominant peptide A2 either alone or in the presence of 20*S*(OH) D3 (10^−7^ M). After 72 h supernatants were analyzed by a multiplexed ELISA to determine concentrations of selected cytokines. Concentrations of cytokines are pg/mL and represent the means of three separate analyses. Comparing pep A2 alone to pep A2 plus D3, *p* ≤ 0.01 for IL-4, IL-10 and IFN-γ and *p* ≤ 0.001 for IL-17.

**Figure 3 ijms-22-13342-f003:**
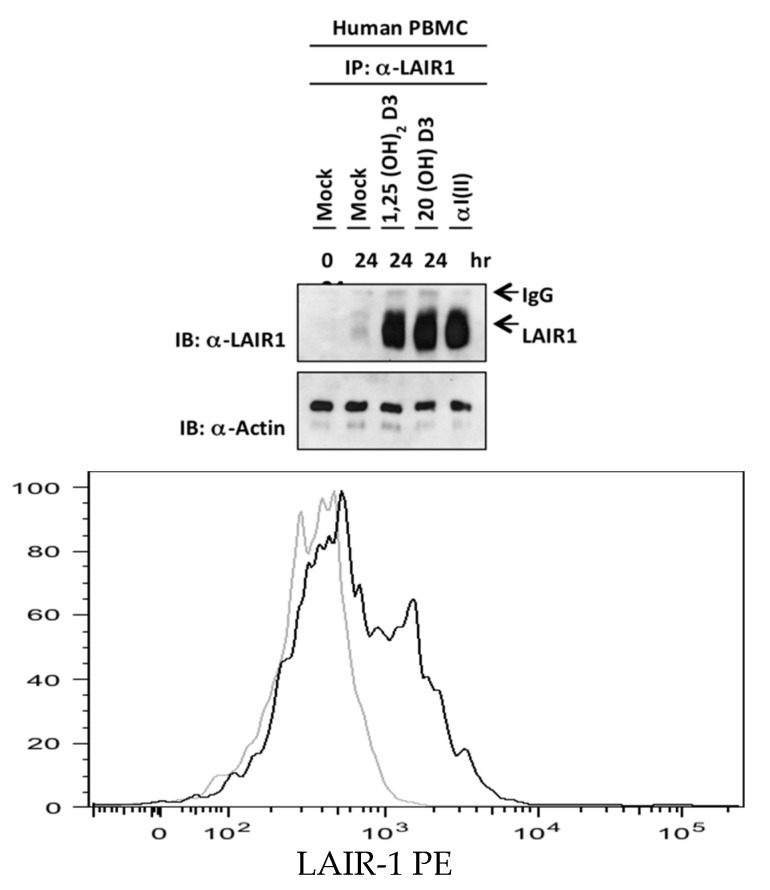
Stimulation of the expression of LAIR-1 by active forms of vitamin D3. (Upper panel) Human PBMCs from normal volunteers were activated by Mock control (ethanol, 10^−8^ M), 1,25(OH)_2_D3 (10^−8^ M), 20*S*(OH)D3 (10^−8^ M), or αI(II) (100 μg/mL) for 12 h. Total proteins (400 μg/500 μL reaction volume) were immunoprecipitated with protein A/G bead conjugated with anti-LAIR-1 (R&D Systems) antibody. Expression of LAIR-1 was examined by Western blot analysis using the anti-LAIR-1 antibody. Actin was used as input control. This experiment is representative of three separate experiments. (Lower panel) Mouse splenocytes were cultured 12 h with either vehicle control (ethanol, 10^−8^ mol/L), (light grey line, mean fluorescence [MF] 388 ± 25)), or 1,25(OH)_2_D3 (10^−8^ mol/L) (black line, MF 1469 ± 42), and stained with anti-LAIR-1 for analysis by flow cytometry. The indicated panels were gated for CD4^+^ cells.

**Figure 4 ijms-22-13342-f004:**
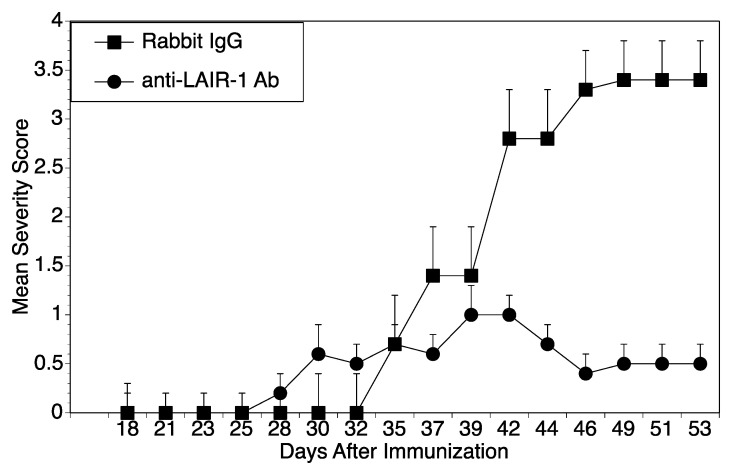
Treatment with Polyclonal antibody to LAIR-1. DR1 mice were immunized with CII/CFA to induce arthritis. Six mice were given intraperitoneal injections on day 0, 7, and 14 with 100 µg/dose of α-LAIR polyclonal IgG antibodies and seven mice were given 100 µg/dose of normal rabbit IgG as a control. Mice were scored for severity. The mice treated with antibody to LAIR-1 had significantly less arthritis severity than mice treated with control beginning on day 42 (*p* ≤ 0.05).

**Figure 5 ijms-22-13342-f005:**
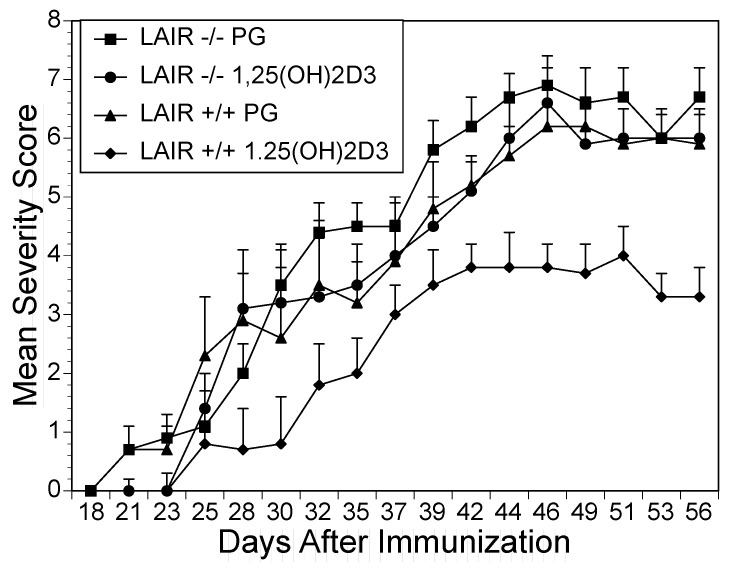
Treatment of LAIR-1^−/−^ mice with 1,25(OH)_2_D3 or 20S(OH)D3. (Upper Pane) Treatment with 1,25(OH)_2_D3. 20 DR1 LAIR-1^+/+^ and 20 DR1 LAIR-1^−/−^ mice, were immunized with CII/CFA to induce an arthritic response and were observed for the development of arthritis. Each LAIR-1^+/+^ and LAIR-1^−/−^ mouse was treated with an oral dose of either vehicle control (propylene glycol, 0.1 mL/dose) or 1,25(OH)_2_D3 (23 μg/dose). The mice were scored visually for severity of disease and the mean severity of arthritis scores for each group were calculated. (Lower Panel) *Treatment with 20S(OH)D3*. 10 DR1 LAIR-1^+/+^ and 10 DR1 LAIR-1^−/−^ mice, were immunized with CII/CFA to induce arthritis. Beginning on day 13 and continuing to day 48, each LAIR-1^+/+^ and LAIR-1^−/−^ mouse was treated with an oral dose of either vehicle control (propylene glycol, 0.1 mL/dose) or 20S(OH)D3 (15 μg/kg/dose). The mice were scored visually for severity of disease and the mean severity of arthritis scores for each group were calculated. Beginning on day 44 (*p* < 0.05) for LAIR-1^+/+^ mice that received either 1,25(OH)_2_D3 or 20*S*(OH)D3 supplementation compared to LAIR-1^+/+^ mice fed PG.

**Table 1 ijms-22-13342-t001:** Cytokine responses in splenic T cells from mice fed a vitamin D deficient diet. Splenocytes from DBA/1 mice fed either Vitamin D+ or Vitamin D− diet and previously immunized with CII/CFA were cultured with A2 peptide (3 µmol/mL) or no Ag, then supernatants were analyzed for cytokines (*n* = 3 for each group). Values indicated represent the mean ± SEM (pg/mL) of three separate experiments. * *p* ≤ 0.001.

		IFN-γ	IL-17A	IL-10	IL-4
Vitamin D+	No Ag	217 ± 20	158 ± 24	5 ± 2	2 ± 6
	A2 Peptide	1143 ± 77	2128 ± 181	53 ± 7	5 ± 2
Vitamin D−	No Ag	180 ± 17	127 ± 25	4 ± 3	2 ± 4
	A2 Peptide	14,806 ± 70 *	8685 ± 220 *	2095 ± 17 *	35 ± 15 *

Cytokines pg/mL.

## Data Availability

The datasets used and/or analyzed during the current study are available from the corresponding author on reasonable request.

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
