# Peer review of "1,25-Dihydroxyvitamin D3 and 20-Hydroxyvitamin D3 Upregulate LAIR-1 and Attenuate Collagen Induced Arthritis"

_ijms, 2021, doi:10.3390/ijms222413342_

Round 1
Reviewer 1 Report
In this study the authors investigated the activity of both 1,25(OH)2D3 and 20S(OH)D3 in Rheumatoid Arthritis (RA). In a collagen induced arthritis (CIA) model, mice fed a diet deficient in vitamin D develop a more severe arthritis. Vitamin D upregulates LAIR-1 and the activated LAIR-1 decreases RA signs. Vitamin D influences RA by LAIR activity, demonstrated in LAIR-1 KO animals, in which vitamin D loses its efficacy.
The study is interesting and is a starting point for further studies employing 20S(OH)D3 as immunomodulator.
I find interesting that 20S(OH)D3, given at lower doses than 1,25(OH)2D3, decreases severity score more potently than 1,25(OH)2D3, this could be highlighted in text.
The study is well performed in general, although some details perhaps need revision or clarification.
- Figure 2: 1,25(OH)2D3 has not been tested; Figure 3 lower panel: 20S(OH)D3 has not been tested.
- It is not clear why in splenocytes all measured cytokines are downregulated if the diet contains vitamin D compared to vitamin D deficient diet (table 1) but when the authors treat the same cells with 20S(OH)D3 the result is not the same. Moreover, the treatment with 1,25(OH)2D3 is missing.
- Figure 5: in results, the authors state that “the data also show that LAIR-1-/- mice had severity scores no different than the untreated wild type controls irregardless of 1,25(OH)2D3 treatment”. This conclusion is not so convincing referred to lower panel. Revise the text or if the difference is not statistically significant, please specify in the legend. If LAIR-1 activity is so impacting in RA, then why LAIR-1 KO does not increase severity of RA? The animal model should have high expression of LAIR-1, similarly to cells treated in vitro with αI(II) (figure 3), if this is not the case please explain.
The legend of figure 5 also contains conclusions about the meaning of the data, which should be moved to the text and not included in legend, to simplify the legend. This applies to all legends, if you like.
Author Response
Reviewer #1
I find interesting that 20S(OH)D3, given at lower doses than 1,25(OH)2D3, decreases severity score more potently than 1,25(OH)2D3, this could be highlighted in text.
Response: This sentence has been added to the text as the reviewer requested.
The study is well performed in general, although some details perhaps need revision or clarification.
- Figure 2: 1,25(OH)2D3 has not been tested; Figure 3 lower panel: 20S(OH)D3 has not been tested.
Response: A reference giving T cell cytokine results following treatment of spleen cells with 1,25(OH)2D3 has been added to the results section attached to Figure 2. Although 20S(OH)D3 was not tested by flow cytometry in the lower panel of figure 3, the western blot in the upper panel of figure 3 shows clearly that LAIR-1 is upregulated following culture with 20(OH)D3.
- It is not clear why in splenocytes all measured cytokines are downregulated if the diet contains vitamin D compared to vitamin D deficient diet (table 1) but when the authors treat the same cells with 20S(OH)D3 the result is not the same. Moreover, the treatment with 1,25(OH)2D3 is missing.
Response: The increase in Th2 cytokine responses observed following treatment with 20(OH)D3 is similar to observations made following treatment with 1,25 (OH)2D3, yet differs from the downregulation of all T cell cytokines noted in Table I in cells from mice given vitamin D sufficient diets. Although both conditions induce a relative shift away from an inflammatory T cell profile, diets sufficient in vitamin D expose the entire immune system of the animal to vitamin D, modifying a large number of T cells over several months. Treatment of normal cells with 20(OH)D3 in vitro reflects a much shorter exposure. We believe these varying conditions, as well as differences in the shifting B/T cell rations that vitamin D induces explain the differing results. This explanation has been added to the manuscript and a reference showing data from treatment with 1,25(OH)2D3 has been added.
- Figure 5: in results, the authors state that “the data also show that LAIR-1-/- mice had severity scores no different than the untreated wild type controls irregardless of 1,25(OH)2D3 treatment”. This conclusion is not so convincing referred to lower panel. Revise the text or if the difference is not statistically significant, please specify in the legend. If LAIR-1 activity is so impacting in RA, then why LAIR-1 KO does not increase severity of RA? The animal model should have high expression of LAIR-1, similarly to cells treated in vitro with αI(II) (figure 3), if this is not the case please explain.
Response: The text has been modified to more accurately reflect the data as the reviewer requested. The reviewer is correct that the LAIR-1 KO mice have more severe arthritis than wild type controls. This is reflected most clearly in the lower panel of figure 5.
The legend of figure 5 also contains conclusions about the meaning of the data, which should be moved to the text and not included in legend, to simplify the legend. This applies to all legends, if you like.
Response: All descriptions of results have been significantly shortened in figure legends except when needed to include p values.
Reviewer 2 Report
Main observations:
- The chosen study design is appropriate.
- The manuscript topic is consistent with the journal content.
- The authors conclude (based on an animal model) that the role of vitamin D on inflammation is at least partially mediated by LAIR-1 and that non-calcemic 20S(OH)D3 may be a promising therapeutic agent for the treatment of autoimmune diseases such as rheumatoid arthritis.
4.The contribution of the manuscript to the topic of the paper can be considered limited.
- The discussion is consistent with the evidence and arguments presented and addresses the stated primary objective.
- It is recommended that the entire manuscript and literature be reviewed for spelling errors.
- In my opinion, this study would be a candidate for publication in your journal as an original article, with minor revisions.
Please use actual units - instead of "M" it should be "mol/l" - for example, in the sentences: “A buffer was pre-pared containing 20mM HEPES (pH7.4), 100 mM NaCl, 0.1mM dithiothreitol, 2μ M human cytochrome P450 scc, 0.1mM EDTA, 0.3μ M adrenodoxin reductase, 10μ M adrendoxin, 2mM glucose 6-phosphate, 2U/ml glucose 6-phosphate dehydrogenase, and 50μ M NADPH.” AND “The PBMCs were set up in culture overnight with vehicle control (ethanol, 10-8M), 1,25(OH)2D3 (10-8 M), 20S(OH)D3 (10-8 M), or αI(II) (100 μg/ml) overnight.”
Lack of LIMITATION section (at the end of the Discussion section).
Lack of the name of the statistical program and the name of the test for checking normality of date distributions – please introduce.
Literature is relatively out of date - more than 80% are articles more than 5 years old - more current items should be used.
Need to check BIBLIOGRAPHY. Double numbering of bibliography (1-56 and 1-57) makes it difficult to assess correctness.
The information about the effect of 1,25(OH)2D3 on more than 900 genes is greatly underestimated - (please check more recent information - not - from 2013 - (Martin Kongsbak, Trine B. Levring, Carsten Geisler and Marina Rode von Essen*The vitamin D receptor and T cell function. Front. Immunol., June 18, 2013 | https://doi.org/10.3389/fimmu.2013.00148) as the bibliographic entry for the sentence: “The binding of 1,25(OH)2D3 to the intracellular VDR regulates more than 900 genes involved in many physiological processes [19].”
Minor comments:
Instead of:
Kongsbak, M.; Levring, T. B.; Geisler, C.; von Essen, M. R., The vitamin d receptor and T cell function. Frontiers in immunology 2013, 4, 148.
Should read:
Kongsbak, M.; Levring, T. B.; Geisler, C.; von Essen, M. R., The vitamin D receptor and T cell function. Frontiers in immunology 2013, 4, 148.
Instead:
Treated mice fed a Vit D- diet were significantly different from controls starting at day 39 (p ≤0.05 using Mann and Whitney analysis).
Should read:
The treated mice fed the Vit D- diet were significantly different from controls beginning at day 39 (p ≤0.05 using Mann-Whitney analysis).
Author Response
Reviewer #2
- It is recommended that the entire manuscript and literature be reviewed for spelling errors.
Response: The manuscript has been reviewed for spelling errors as requested.
- In my opinion, this study would be a candidate for publication in your journal as an original article, with minor revisions.
Please use actual units - instead of "M" it should be "mol/l" - for example, in the sentences: “A buffer was prepared containing 20mM HEPES (pH7.4), 100 mM NaCl, 0.1mM dithiothreitol, 2μ M human cytochrome P450 scc, 0.1mM EDTA, 0.3μ M adrenodoxin reductase, 10μ M adrendoxin, 2mM glucose 6-phosphate, 2U/ml glucose 6-phosphate dehydrogenase, and 50μ M NADPH.” AND “The PBMCs were set up in culture overnight with vehicle control (ethanol, 10-8M), 1,25(OH)2D3 (10-8 M), 20S(OH)D3 (10-8 M), or αI(II) (100 μg/ml) overnight.”
Response: The text has been modified as request to use “mol/l”.
Lack of LIMITATION section (at the end of the Discussion section).
Response: A limitations section has been added at the end of the Discussion as the reviewer requested.
Lack of the name of the statistical program and the name of the test for checking normality of date distributions – please introduce.
Response: The names of the statistical programs used and the tests used for checking normality of data distributions has been added to the Materials and Methods section.
Literature is relatively out of date - more than 80% are articles more than 5 years old - more current items should be used.
Response: Newer references have been added to cover the important topics addressed in this manuscript as the reviewers requested.
Need to check BIBLIOGRAPHY. Double numbering of bibliography (1-56 and 1-57) makes it difficult to assess correctness.
Response: The extra bibliography has been removed as requested.
The information about the effect of 1,25(OH)2D3 on more than 900 genes is greatly underestimated - (please check more recent information - not - from 2013 –
(Martin Kongsbak, Trine B. Levring, Carsten Geisler and Marina Rode von Essen*The vitamin D receptor and T cell function. Front. Immunol., June 18, 2013 | https://doi.org/10.3389/fimmu.2013.00148) as the bibliographic entry for the sentence: “The binding of 1,25(OH)2D3 to the intracellular VDR regulates more than 900 genes involved in many physiological processes [19].”
Response: A new sentence reflecting greater gene involvement for vitamin D as well as a newer reference has been added as the reviewer requested.
Minor comments:
Instead of:
Kongsbak, M.; Levring, T. B.; Geisler, C.; von Essen, M. R., The vitamin d receptor and T cell function. Frontiers in immunology 2013, 4, 148.
Should read:
Kongsbak, M.; Levring, T. B.; Geisler, C.; von Essen, M. R., The vitamin D receptor and T cell function. Frontiers in immunology 2013, 4, 148.
Response: The reference has been corrected as requested.
Instead:
Treated mice fed a Vit D- diet were significantly different from controls starting at day 39 (p ≤0.05 using Mann and Whitney analysis).
Should read:
The treated mice fed the Vit D- diet were significantly different from controls beginning at day 39 (p ≤0.05 using Mann-Whitney analysis).
Response: The sentence has been corrected as requested.
Round 2
Reviewer 1 Report
The article has been revised and is now ready for publication